# Gastric Leptin and Tumorigenesis: Beyond Obesity

**DOI:** 10.3390/ijms20112622

**Published:** 2019-05-28

**Authors:** Kyoko Inagaki-Ohara

**Affiliations:** Division of Host Defense, Department of Life Sciences, Faculty of Life and Environmental Sciences, Prefectural University of Hiroshima, 5562 Nanatsuka, Shobara, Hiroshima 727-0023, Japan; k-inagaki@pu-hiroshima.ac.jp; Tel.: +81-824-74-1795; Fax: +81-824-74-0191

**Keywords:** leptin, carcinogensis, stomach, obesity

## Abstract

Leptin, an adipocyte-derived hormone and its receptor (ObR) expressed in the hypothalamus are well known as an essential regulator of appetite and energy expenditure. Obesity induces abundant leptin production, however, reduced sensitivity to leptin leads to the development of metabolic disorders, so called leptin resistance. The stomach has been identified as an organ that simultaneously expresses leptin and ObR. Accumulating evidence has shown gastric leptin to perform diverse functions, such as those in nutrient absorption and carcinogenesis in the gastrointestinal system, independent of its well-known role in appetite regulation and obesity. Overexpression of leptin and phosphorylated ObR is implicated in gastric cancer in humans and in murine model, and diet-induced obesity causes precancerous lesions in the stomach in mice. While the underlying pathomechanisms remain unclear, leptin signaling can affect gastric mucosal milieu. In this review, we focus on the significant role of the gastric leptin signaling in neoplasia and tumorigenesis in stomach in the context of hereditary and diet-induced obesity.

## 1. Introduction

Humans have been exposed to malnutrition due to starvation. However, recently, malnutrition in the form of over-nutrition has become a serious social problem, because diet-induced obesity due to an increase in nutrient intake and insufficient energetic expenditure, has dramatically increased worldwide and is implicated in numerous metabolic disorders, including cardiovascular diseases and type 2 diabetes (T2D). Furthermore, obesity is a critical risk factor for cancers in various tissues. Prevalence of obesity has been steadily increasing over the past a few decades in not only developed, but also in developing countries, shaping up as a worldwide health problem for children to adult.

Adipocyte-derived leptin, discovered in 1994, modulates appetite and body weight. However, in the case of leptin resistance in obese patients, despite the high concentrations of leptin in blood, appetite is not suppressed, due to decreased sensitivity of its receptor signaling [1,2]. Gastric leptin was discovered in 1998, has been shown to accelerate the absorption of nutrients, although its significance is still being investigated. Leptin exerts pleiotropic effects on immunity and cell differentiation/proliferation in the physiological state [3]. Leptin is mainly produced by adipocytes, besides other tissues, including the stomach [4], and placenta [5]. Leptin secreted by such tissues seems to be independent of the satiety regulation. However, overexpression of leptin and activation of ObR has been reported in various cancers, such as those of the stomach and mammary gland, and is mediated by inflammation, angiogenesis, stemness [6], and epithelial–mesenchymal transition (EMT) and progression [7] (Figure 1).

Gastric cancer (GC) is the fourth most common malignancy and the second leading cause of cancer-related deaths [8,9]. There are critical geographical differences regarding its incidence, mortality, social, culture, and economic entities. More than 100,000 new cases were diagnosed in the world in the last 10 years [10,11]. In Europe, 140,000 individuals were diagnosed in 2014 and 107,000 patients died that same year [12]. One of the reasons for the high mortality in GC patients is the lack of a biomarker for diagnosis at the early stage of GC, besides the high relapse rate after tumor resection. *Helicobacter pylori* (*H. pylori*) infection is well known as a causative factor of GC. through an initiation of epigenetics reprogramming of host cells [13] and dysregulation of β-defensins, antimicrobial peptides [14]. However, *H. pylori*-infected individuals constitute approximately half of the world population, with only a small portion of them eventually developing GC. Recently development of gastric cardia adenocarcinoma has been reported to be strongly linked to obesity [15,16], suggesting the possibility of causes other than *H. pylori* infection. In this review, we focus on the role of leptin as an inducer of carcinogenesis and its implications in GC.

## 2. Background of Leptin and Its Receptor Signaling

Leptin is a 16-kD non-glycosylated protein, consisting of a 167-amino acid polypeptide with 21-amino acids signal sequence at the amino-terminus, which is cleaved following the translocation of leptin into microsomes and is then secreted into the blood stream. The significance of leptin was first discovered by parabiosis experiments. Mutant mice (*ob/ob*) were found to exhibit severe obesity with decreased basic metabolism and physical activity [17], whereas crossing them with wild-type mice led to restoration of normal body weight [18]. Later, *ob* gene was identified to be response for obesity in *ob/ob* mice and subsequently named ‘leptin’, which is derived from a Greek word, ‘leptos’ (‘thin’) [1]. Leptin receptor (ObR) encoded by the *db* gene in mice, is a member of class I cytokine receptor family, similar to gp130 [3,19,20]. ObR consists of six isoforms (ObRa, ObRb, ObRc, ObRd, ObRe and ObRf), formed from alternative RNA splicing of the *db* gene. These isoforms have a common leptin-binding domain, and differ in their intracellular domains [21,22]. Among them, ObRb, named as the long form, is the only isoform that contains the intracellular motif necessary for the leptin-mediated JAK-STAT pathway, whereas four short forms (ObRa, c, d, and f), differ in their cytosolic carboxy terminals. ObRe is a soluble form that lacks the trans-membrane domain and is involved in blood to brain leptin transport by antagonizing leptin endocytosis [23]. The extracellular domain of the ObRb is constituted of homology region 2 (CHR2), which contains flexible hinge regions [24]. Upon leptin binding, ObRb dimerizes, resulting in the activation and tyrosine (Y) phosphorylation of JAK2, subsequently forming PY-985 and PY-1138 in ObRb. PY-1138 of ObRb recruits the SH2 domain-containing transcription factor STAT3, resulting in tyrosine phosphorylation of STAT3 and its translocation to the nucleus [25]. In addition, PY-1077 promotes the recruitment and activation of STAT5 [26]. The lack of STAT5 in the central nervous system causes to obesity with hyperphagia, whereas STAT5 activation in hypothalamic neurons suppresses food intake [27], suggesting that not only STAT3, but the JAK2-STAT5 pathway also contributes to the prevention of obesity. PY-985, to which SOCS3 binds to recruit the SH2 domain-containing protein-tyrosine phosphatase SHP-2, functions upstream of extracellular signal regulated kinase (ERK) activation and c-fos transcription [28]. Once inside the nucleus, STAT3 mediates gene transcription, including transcription of the suppressor of cytokine signaling, SOCS3, which in turn binds to PY-985 and JAK2 to attenuate leptin receptor signaling (Figure 2).

## 3. Adipocyte-Derived Leptin in the Central Nervous System

Leptin secreted by adipose tissue reaches the hypothalamus by receptor-mediated transcytosis rather than by the blood-brain barrier system [29]. ObRb is located in the arcuate nucleus of the hypothalamus, which is responsible for the secretion of neuropeptides and neurotransmitters that suppress appetite and body weight [2,21]. Further studies on gene-targeted mice have revealed that leptin’s action in the central nervous system only is sufficient to regulate body weight, feeding, energy expenditure, glucose metabolism, and behavior [30]. However, leptin’s anorexigenic effects are suppressed in obese individuals and high-fat diet-induced obese animals, despite the elevated serum leptin levels is; when leptin signaling does not function, a condition termed “leptin resistance” [31,32,33,34,35]. In humans, congenital mutation of the *ob* gene is rare, although patients with early-onset extreme obesity have been reported; cousins from a consanguineous Pakistani family, involved the absence of leptin due to homozygous frame shift mutation in leptin gene [36], and a boy of Turkish consanguinity, whose parents were of normal weight was reported to be obese due to homozygous transversion (c.298G -> T), resulting in mutated leptin that failed to bind ObR [37].

SOCS3 has been proposed as a potential mediator of leptin resistance. Peripheral administration of leptin to *ob/ob* mice was found to specifically induce SOCS3 mRNA in regions of the hypothalamus that are known to be important for feeding behavior [38]. Mice with SOCS3 deletion either in the whole brain or in proopiomelanocortin (POMC) neurons (the key leptin target neurons in the arcuate nucleus of the hypothalamus) are resistant to high-fat diet-induced obesity [39,40]. Furthermore, studies using either POMC or ObRb-expressed neuron-specific SOCS3 transgenic mice indicated that SOCS3 upregulation alone in proopiomelanocortin (POMC) neurons, but not in ObRb neurons, is sufficient to cause leptin resistance and obesity, mediated by the antagonization of phosphorylated STAT3 and mTOR-S6K signaling due to SOCS3 upregulation [41].

## 4. Gastric Leptin as a Causative Factor of Carcinogenesis in Stomach

In the gastrointestinal system, leptin is constitutively produced in the stomach, by parietal cells that secrete gastric acid, and chief cells that produce pepsinogen and gastric lipase, but not in bowels, whereas ObR is expressed in both the stomach and bowels [4,42,43]. Thus, the stomach is a unique organ, which constitutively expresses both leptin and ObR, and can transduce autocrine leptin signaling. Two main effects of gastric leptin may be considered: First, its effect on intestinal function. Leptin is secreted in the gastric lumen and enters the intestine to directly act on the intestinal epithelium to enhance absorption of peptides by Pept-1, glucose and galactose by Glut-5 in the physiological state [44,45], promotes cell invasive capacity via the PI3K, Rho- and Rac-dependent pathway in colon cells [46], along with cell proliferation [47], which is beneficial effect for both normal and repaired mucosa such as that after inflammation. Leptin-deficient *ob/ob* mice, which underwent resection of the small intestine, showed reduced cell proliferation and enhanced apoptosis compared to lean mice that received the same treatment [48]; however, contradictory reports showed increased apoptosis and accelerated intestinal adaptation [49]. Unlike leptin signaling in the hypothalamus, that in the gut is independent of the obesity development. Intestinal epithelium specific-ObR-deficient mice using villin-promoter are not spontaneously obese [50], implying that intestinal leptin signaling is independent of the metabolism and appetite. Rather, villin-ObRb conditional KO (cKO) mice showed a reduced number of aberrant crypt foci and colon tumors induced by azoxymethane, mediated through activation of ObRb-STAT3 signaling [51], thus indicating that intestinal leptin signaling controls the development and cell fate of epithelial cell carcinogenesis. In terms of intestinal infection, *Entamoeba histolytica*, which is an anaerobic parasitic amoebozoan, induces severe diarrheal diseases. Both *db/db* mice and villin-ObR cKO mice are susceptible to *E. histolytica* infection to exhibit severe mucosal destruction of the intestine, indicating leptin signal to be an essential factor for host resistance against amebiasis [52]. Leptin acts as a pro-inflammatory cytokine that promotes the Th1 immune response, and Th1-type cytokines, in particular IFN-γ are critical for the protection against *E. histolytica*. Both *db/db* mice and villin-ObR cKO mice were unable to transduce leptin signaling to facilitate higher susceptibility to *E. histolytica*. Second, the effect of gastric leptin on gastric function is still an enigma, although both leptin and ObR are expressed in parietal cells and chief cells [4,53,54]. First report of enhanced gastric leptin was in GC caused by *Helicobacter pylori* infection [55]. *H. pylori* infection significantly increased in gastric leptin levels while its cure decreased the levels, whereas serum leptin remained unaffected by *H. pylori*. Leptin stimulates cell proliferation, and its effect on the development and progression of GC is seeded in GC cells, including advanced GC cells [56,57,58]; expression of gastric leptin might be a prognostic marker in poorly differentiated GC [59]. Lee et al. had demonstrated that both leptin and ObR are overexpressed in gastric adenoma, and in early or advanced GC, and that leptin activates STAT3-ERK1/2 pathway leading to VEGF expression in GC cell line [60]. *In vitro*, leptin induces the migration of AGS cell (adenocarninoma) and MKN-45 cell (poorly differentiated adenocarcinoma) by upregulating ICAM-1 expression [61], hence suggesting that leptin can activate or modulate a variety of signaling cascades. We demonstrated that overexpression of leptin and its signaling was stimulated by SOCS3 deletion in gastrointestinal epithelium-specific using T3b-promoter [62]. T3b-SOCS3 cKO mice showed spontaneous gastric tumor by enhancing the ObRb-STAT3 pathway. Although T3b-SOCS3 cKO mice underwent SOCS3 deletion in both small and large intestines as well as in stomach, the tumor was restricted in the stomach. All T3b-SOCS3 cKO mice died within half year. The cKO mice exhibited hyperplasia at 3 weeks of age and developed tumors by 8 weeks, indicating that leptin can stimulate rapid tumorigenesis of epithelial cells in the stomach. This was supported by excess of leptin and ObR being associated with tumorigenesis in patients with early stage of GC [58]. As another unique point of this model, gastric tumor formation occurs prior to the inflammatory lesions. Chronic inflammation is considered to raise the incidence of cancers in various tissues. Gp130 is ubiquitously expressed and its mutant gp130^757F^ mice developed inflammation-associated gastric adenoma because mutated gp130 cannot bind to SOCS3 with decreased in leptin expression in the gastric mucosa [63]. In contrast, in T3b-SOCS3 cKO mice, CD45^+^ infiltrated cells clearly appeared at 15 weeks of age, when gastric tumor formation was already completed. The evidence demonstrated that leptin is a trigger that stimulates malignancy and carcinogenesis; in particular, the stomach possibly prompts autocrine leptin signaling. Tumor-initiating stem cells have been reported to potently express ObR, resulting in tumor progression mediated by the activation of STAT3 and induction of pluripotency-associated transcription factors, such as Oct4 and Sox2 [64]. This evidence implied that leptin actively affects tumor formation and supports cell proliferation and pluripotency for tumorigenesis.

Murine models of GC, which closely mimic human GC pathology, have enriched our understanding of the disease (Table 1), either chemical carcinogen-induced or by bacterial infection, or using gene-targeting model focused on gastric function. N-methyl-N-nitrosourea (MNU) is an effective chemical for carcinogenesis, although limited to the antral region, and rarely in fundus [65,66]. *H. pylori* is considered to account for major of gastric cancer, however, human *H. pylori* infection is difficult to induce in murine models, with the exception of Mongolian gerbils. While CagA is a virulence factor of *H. pylori*, only a small portion of *H/K-ATPase* promoter-driven *cagA* Tg mice develop adenocarinoma after 72 weeks [67]. Thus, *H. felis* is frequently used as an alternative for murine model of gastric *Helicobacter* infection. Moreover, molecular-based approach using gene-targeting models has revealed a variety of processed of tumorigenesis [68,69]. TFF1 is a mucin-associated peptide secreted by gastric pit and neck cells, and found to be frequently downregulation in gastric carcinoma [70,71]. Gastrin is released by G cells of the stomach, and stimulates the secretion of gastric acids by the parietal cells. *ATP4a* and *ATP4β* genes encode α and *β* subunits of H^+^K^+^ATPase, which are responsible for the secretion of gastric acid. Despite these molecules being critical for the maintenance of normal functioning of gastric mucosa, mice with TFF1, gastrin or ATP4a deleted developed precancerous lesions or incomplete cancer [72,73]. K-ras has been shown to be strongly linked to the development of human cancers. Ubiquitous K-ras activation rapidly induces hyperplasia, metaplasia and adenomas in the stomach, while carcinogenesis has been observed in other tissues such as the oral cavity, colon, liver and lung [74]. In terms of cytokine signaling, deletion of TGF-β1 caused 40% incidence of adenocarcinoma [75], and lack of its downstream molecules Smad3 and Smad4 caused carcinoma, based on E-cadherin downregulation and invasive neoplasia [76,77]. Tgfβ1-C33S crossed with Rag2^−/−^ chimeric mice exhibited reduced inflammation and tumors, implying that TGF-β1-mediated immune responses can suppress tumor development. IL-1β is strong association with increased risk of GC, whereas ATP4b prompter-driven IL-1β Tg mice exhibited only 30% incidence of adenocarcinoma [78]. These studies using gene-targeted mice, collectively indicate that mice can rarely develop perfectly spontaneous GC at young age, and that leptin signaling in the stomach can potently and rapidly induce GC.

## 5. Gastric Neoplasia Triggered by Diet-Induced Obesity

Obesity is one of the causative factors of GC. Excessive leptin secretion/augmentation of leptin signaling occurs in HFD-feeding. We demonstrated that high-fat diet (HFD) induces the loss of parietal cells and glandular cells, and promotes intestinal metaplasia, which transforms the gastric epithelium to an intestine-like epithelium, or precancerous lesions [85]. The stomach is so sensitive to HFD that 1week of HFD feeding resulted in hyperplasia. At 3 months of feeding, complete loss of zymogenic and glandular metaplasia occurred, and remarkable nuclear atypia such as nuclear elongation and pseudostratification was seen at 12w and no normal gastric gland and at 20 weeks. HFD induces ectopic expression of the following in the stomach: Muc2, an intestinal type mucus protein; Cdx2, a transcription factor that directs development of the intestinal epithelium; PLA2, a Paneth cell marker; in contrast, it causes the loss of H^+^K^+^ATPase, a parietal cell marker and integral membrane protein responsible for gastric acid secretion. Interestingly, these changes similar to those in carcinogenesis, were suppressed in *db/db* mice. The *db/db* mice are extraordinarily obese, yet showed suppressed pathogenesis.

HFD feeding also induces ectopic expression of fat in the stomach [86]. Accumulating ectopic fat can cause cell injury and functional impairment in a variety of tissues, also known as ‘lypotoxicity’ [87]. In terms of localization, fat accumulation was observed in the cytosol, not in the nucleus. Such a state of lipid stress in non-adipose tissues affects organelle homeostasis. Examining lysosome, mitochondria, Golgi apparatus, and endoplasmic reticulum, we found LAMP2A, a lysosomal marker protein, to be elevated in the early period after HFD feeding [86]. LAMP2A is a lysosome-associated membrane protein, which acts as a receptor for the substrates of chaperon-mediated autophagy, and is highly expressed in gastric cancer [88]. In addition, acidic lipase, which is produced by the chief cells in the gastric fundus and functions in the lysosome to maintain low pH of the gastric mucosal milieu [89], is ectopically expressed in the duodenum. In contrast, lipoprotein lipase, which is a classical lipase produced and secreted by pancreas into the duodenum, is expressed and in the gastric mucosa. HFD might induce not only intestinal metaplasia, but also gastric metaplasia. Recently, accumulating evidence has revealed the link between lysosomal dysfunction and obesity. HFD downregulates autophagy by reducing autophagosomes and lowering acidity of lysosomes both *in vitro* and *in vivo* [90,91]. Although HFD-related lipid changes have been reported to reduce LAMP2A expression in neurodegenerative diseases [92], earliest changes of LAMP2A may be considered as one of the host defense mechanism occurring in the stomach. Following LAMP2A change, COX IV-2 expression was increased, which is the hypoxia marker molecule in the mitochondria of the stomach by HFD feeding. Hypoxia is a common condition in most of tumors, referring to a non-physiological oxygen level, leading to the acquisition of EMT [93]. Hypoxia-inducible factor 1α (HIF-1α) is a master regulator of adaptive responses to hypoxia and promotes critical steps in tumor progression [94] as well as obesity [95]. HIF-1α occurs downstream of STAT3 and regulates Hes-1 expression [96]. Hes-1 belongs to helix-loop-helix family of the transcription factors and regulates the differentiation of stem/progenitor cells to absorptive cells in intestinal tract or in tumor cells, in addition to the normal cells of gastrointestine [97]. Hes-1 is highly expressed in the gastric mucosa, showing intestinal metaplasia induced by HFD feeding [86]. Thus, excessive leptin signaling might be involved in neoplasia and metaplasia in the stomach, mediated by the STAT3-HIF-1-Hes-1 axis. In triple-negative breast cancer, leptin-dependent mechanisms, which lead to cancer, mediated by the upregulation of stem cell (CSC)- and EMT-related gene expression [98], induces estrogen receptor-α (ER-α) expression via JAK-STAT pathway [99]. Leptin-induced ZEB-1, an EMT-induced transcription factor via activation of ERK signaling, was seen to be promoted in lung cancer [100].

β-Catenin is a multi-functional molecule with a dual role. First, it is a cytoplasmic anchoring protein of E-cadherin, required for cell-cell junction as adherens junctions [101]. Secondly, it acts as an intracellular signal transducer in Wnt pathway [102,103]. Target molecules of β-catenin include stem cell and CSC markers, namely Lgr5, CD44, EpCAM [104], Nanog, Oct4, and c-Myc [105]. Suppression of E-cadherin expression and nuclear accumulation of β-catenin are correlated with tumor progression in many cancers including GC [103]. In murine models with upregulation of leptin signaling in the stomach, β-catenin localization was clearly shown; the gastric mucosa due to SOCS3 deletion showed completed nuclear accumulation of β-catenin [62], and HFD-fed gastric mucosa exhibited accumulated intracellular or perinuclear localization [86]. The alteration was observed in an early stage of the pathogenesis (Figure 3).

## 6. Therapeutic Implications

The rising incidence of GC and its poor clinical outcome using current therapeutic approaches urge the discovery of new medicinal approaches, since the survival rate is 40-50% after 1 year, and less than 15% over a 5-year duration after initial therapy treatment, due to relapse [106]. Cisplatin and 5-fluorouracil (5-FU), in addition to trastuzumab, have been used as effective treatments for patients with gastro-esophageal adenocarcinoma, with human epidermal growth factor 2 (HER2) [107], a proto-oncogene, being expressed frequently in the proximal stomach compared to that in the distal [108]. In addition, in HER2-negative GC, oxaliplatin-based regimens could improve tolerance effectively. Another drug, ramucirumab, a monoclonal antibody against vascular endothelial growth factor receptor 2, was shown to be effective in patients with GC that had failed or progressed after the first line therapy [109]. In relation to angiogenesis, although these approaches demonstrate success to some extent, treatment-associated toxicity is a pronounced side-effect. Recently, leptin has been reported as a candidate target molecule for GC therapy since it is considered as a tumorigenesis-associated molecule. Thus, the use of leptin antagonist could be a useful therapeutic strategy against cancer. Leptin-expressing GC cells also exhibited chemoresistance [110,111]. Leptin attenuated the effect of 5-FU on apoptosis via downregulation of pro-apoptotic molecules (BAX, Caspase-3) and upregulation of anti-apoptotic molecules (BcL-xL, RIP) in pancreatic cancer [112]. Moreover, inhibition of leptin-induced NLRP3 inflammasome prevented cancer cell proliferation of MCF-7 breast cancer cells [113]. Recently, leptin has been reported to induce antibodies to *H. pylori* in mice vaccination, suggesting the possibility of adjuvant tool in the development of effective vaccine [114]. All these studies encourage the clinical inhibition of leptin and its signaling as a novel therapeutic approach against GC.

## 7. Conclusions

Gastric cancer and obesity are currently considered as major public health concerns worldwide. Leptin potently affects metaplasia and cancer cell proliferation. In most cases, augmentation of leptin and its signaling promotes carcinogenesis at an early stage. Gastric leptin signaling is a critical checkpoint for the onset and development of gastric neoplasia. Thus, silencing of leptin or leptin-associated upregulated molecules may be considered as a therapeutic treatment.

## Figures and Tables

**Figure 1 ijms-20-02622-f001:**
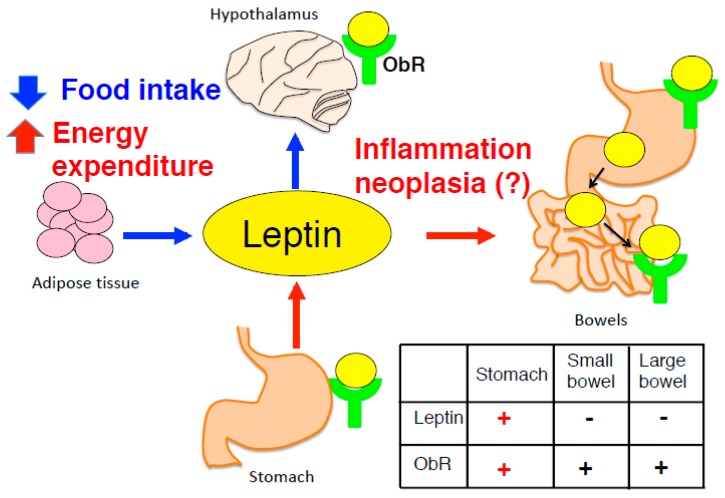
Adipocyte-derived leptin modulates the suppression of appetite and increased energy expenditure mediated by leptin signaling in hypothalamus. By contrast, stomach expresses both leptin and ObR, however, physiological significance of gastric leptin remains unclear. Table inside figure shows expression of leptin and ObR in the gastrointestinal tract.

**Figure 2 ijms-20-02622-f002:**
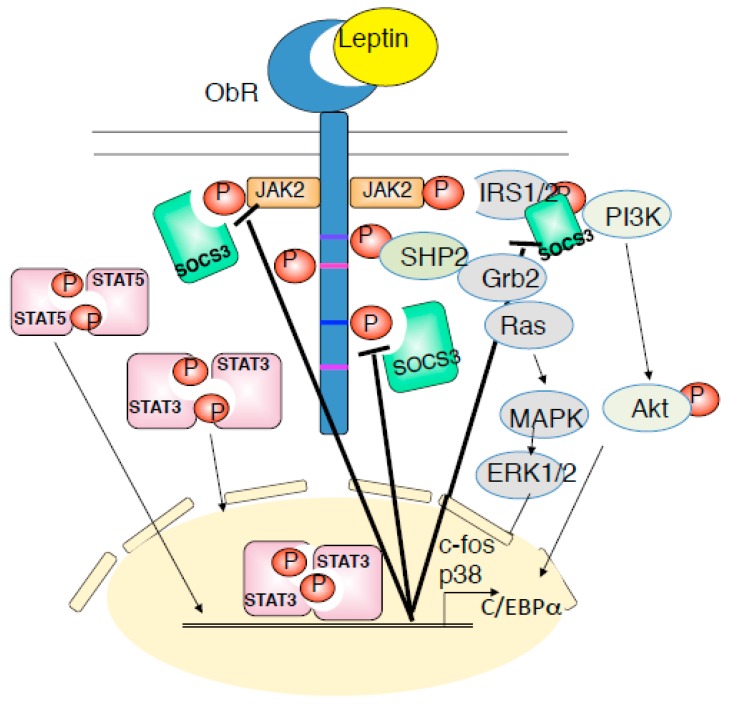
Leptin receptor signaling is mediated by JAK-STAT, PI3K-Akt, and SHP2-ERK pathway. Leptin binds to ObRb and activates JAK2, and induced to the phosphorylation of Tyr985, Tyr1077, and Tyr1138 of ObRb. PY-985, PY-1077, and PY-1138 bind to their downstream molecules and proceeds to phosphorylation of JAK2-STAT3, JAk2-STAT5, PI3K-IRS-Akt, SHP2-ERK pathways. These signaling are negatively regulated by SOCS3. Dysregulation of the leptin receptor signaling is involved in the onset of leptin resistance, inflammation and cancer.

**Figure 3 ijms-20-02622-f003:**
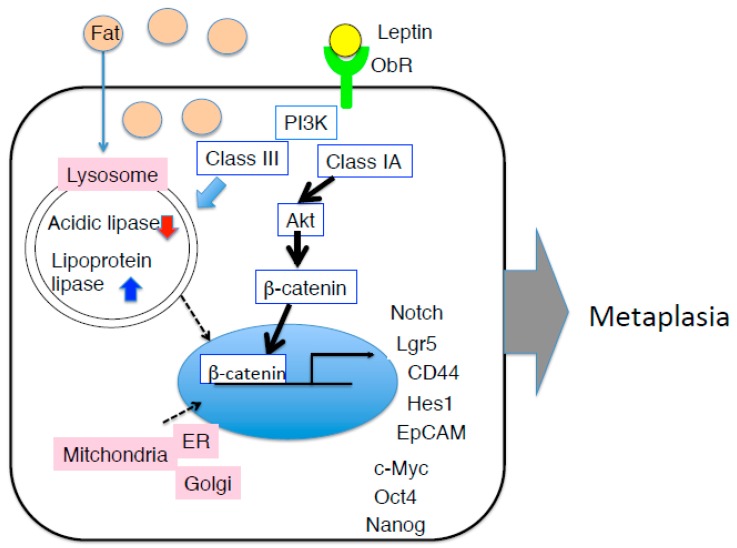
Hypothesis mechanism of intestinal metaplasia of the gastric mucosa in HFD-induced obese mice. In addition to JAK-STAT pathway, by PI3K-Akt-β-catenin pathway may be strongly induce pluripotent gene, leading to the onset of intestinal metaplasia. Amount of HFD causes lipotoxic milieu in the gastric epithelium, in particular, function of lysosome and mitochondria is impaired, although it remains unclear why gastric leptin is increased due to HFD feeding.

**Table 1 ijms-20-02622-t001:** Murine model of gastric cancer.

Model	Incidence (%)	Periods or Age Onset	Phenotype	Causative Mechanism	References No.
**Chemical carcinogen/Bacterial infection induced**					
MNU	(1) 60% (2) 19%	> 50 weeks	Adenocarcinoma	An alkylating reagent in experimental gastric carcinogenesis	(1) [65] (2) [66]
H. felis	80%	< 6months	Severe gastritis	More susceptible Helicobacter spp for C57BL/6 murine model	[68]
MNU + H. felis	40%	9 months	Adenoma		[69]
**Gene-targeting**					
pS2 (TFF1)^−/−^	30%	5 months	Dysplasia	Lacking normal gastric mucus	[71]
Gan (K19-Wnt1/C2mETg)	100%	< 10 months	Carcinoma	Excess of COX-2 and microsomal prostaglandin E synthase-1	[79]
INS-GAS	85%	> 20 months	Intramucosal carcinoma	Hyperexpression of gastrin	[80]
INS-GAS + H. felis	< 8 months				
GAS^−/−^	60%	1 year	Displasia	Lacking gastrin	[72]
Atp4a^−/−^	100%	1 year	Incomplete intestinal metaplasia	Lacking H^+^K^+^ ATPase	[73]
Atp4b-SV40	60%	1 year	Carcinoma, invasion (Lymphatic–vascular), metastasis (liver)	Expression of SV40 in parietal cells	[81]
Atp4b- (CDH1xTrp53)^−/−^	100%	1 year	Metastasized to lymph nodes	Lacking E-cadherin and p53 in parietal cells	[82]
ATP4b-hIL-1b	30%	1 year	Dysplasia, Adenocarcinoma	MDSCs recruitment via IL-1RI/NF-κB pathway	[78]
Kvlqt1^−/−^	100%	3 months	Hyperplasia in gastric neck cells	Lacking potassium channel	[83]
K-ras ^G12D^ (systemic)	100%	< 18 days	Metaplasia	Hyperactivation of MAPK by K-ras mutation	[74]
Tgfβ1-C33S	40%	4-5 months	Well-differentiated adenocarcinomas	Unable forming latent TGF-β binding protein-1	[75]
Smad3^−/−^	100%	10 months	Tumors, invasive neoplasia	Excess of cytosolic E-cadherin	[76]
Smad4^−/−^	100%	> 1 year	Invasive carcinoma	Increased cyclin1 and upregulation of TGF-β1	[77]
RUNX3^−/−^	70%	> 1 year	Hyperplasia, Chief cells loss and increased cdx2	Enhanced Wnt-β catenin signaling by RUNX3 loss	[84]
gp130^757F^	100%	3 months	Adenoma	Abrogating SHP2-Ras-ERK signaling	[63]
T3b-SOCS3^−/−^	100%	2 months	Carcinoma	Augment of leptin expression and ObR-STAT3 signaling by gastrointestinal cell- specific SOCS3 loss	[62]

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
