# Peer review of "Gastric Leptin and Tumorigenesis: Beyond Obesity"

_ijms, 2019, doi:10.3390/ijms20112622_

Reviewer 1 Report

Gastric leptin and tumorigenesis: Beyond the obesity

Kyoko Inagaki-Ohara in the present review focused on the significant role of the gastric leptin signaling in neoplasia and tumorigenesis in stomach in the context of hereditary and diet-induced obesity.

The review is scientifically very interesting, and acceptable after a minor revision. In fact, the Author should add in Paragraph 4 other information on the link among leptin, obesity, gastric cancer and several markers (such as CD10, CD13, CD26, and CD143), to make the review more appealing.

In particular, they should report the findings of some scientific studies such as:

1) Carl-McGrath S et al. The ectopeptidases CD10, CD13, CD26, and CD143 are upregulated in gastric cancer. Int J Oncol. 2004 Nov;25(5):1223-32.

2) Iaffaldano L et al. High aminopeptidase N/CD13 levels characterize human amniotic mesenchymal stem cells and drive their increased adipogenic potential in obese women. Stem Cells Dev. 2013 Aug 15;22(16):2287-97. doi: 10.1089/scd.2012.0499.

3) Nohara S et al. Aminopeptidase N (APN/CD13) as a target molecule for scirrhous gastric cancer. Clin Res Hepatol Gastroenterol. 2016 Sep;40(4):494-503. doi: 10.1016/j.clinre.2015.11.003. Epub 2016 Jan 13.

Author Response

Reviewer #1

I would thank to the Reviewer for noting other target molecules as GC. Among CD10, CD13, CD26, and CD143, CD13 is only possible contributor in both obesity and gastric cancer associated with leptin; CD13 has been reported to increase in the scirrhous gastric cancer and in amniotic mesenchymal stem cells in obese individuals. However, in promotion of cancer stem cell (CSC)-like properties, CD13 intracellular signaling is mediated through neither JAK-STAT, PI3K-Akt, Notch nor Wnt-b-catenin, which are associated with leptin signaling, but providing reducing reactivity to oxygen species (ROS), promoting CSC survival via EMT (Liskova, Molecules 201924(5), 899). In this review, I focus on the role and effect of gastric leptin and its signaling on tumorigenesis, not gastric cancer or obesity themselves. Therefore, considering purport of this review, I cannot add description regarding these molecules in this review, although these are very curious molecules for carcinogenesis. I look forward to new finding which shows the link of these molecules and leptin signaling to develop gastric cancer.

Reviewer 2 Report

The review entitled “Gastric leptin and tumorigenesis: Beyond the obesity” explains the role of leptin as an inducer of carcinogenesis and its  implications in GC.

The data are novel and well defined. The study is significant and appropriately interpreted .

However, I have a few concerns and related suggestions.

The authors did not discuss regarding the several intracellular modifications induced by H. pylori such as epigenetic alterations (Chiariotti et al., 2013) and the dysregulation of defensins (Pero et al., 2017) which are very important and may occur during gastric carcinogenesis.

Author Response

Reviewer #2

I thank to the Reviewer’s concern regarding medications by H. pylori in host cells. We added sentences; “an initiation of epigenetics repropramming of host cells [13]and dysregulation of b-defensins, antimicrobial peptides [14]”(page 4, lines 6-7). As a relevant of H. pyloriinfection, I added sentences, which suggest the possibility of leptin as adjuvant against H. pyloriinfection in section of ‘Therapeutic implications’, because leptin has been reported to induce antibody to respond to H. pylori; Recently, leptin has been reported to induce antibody to H. pylori vaccination in mice, suggesting the possibility of adjuvant tool in the development of effective vaccine [108]”(page 14, lines 19-21).